# The Clinical, Genomic, and Transcriptomic Landscape of BRAF Mutant Cancers

**DOI:** 10.3390/cancers16020445

**Published:** 2024-01-19

**Authors:** Suzanne Kazandjian, Emmanuelle Rousselle, Matthew Dankner, David W. Cescon, Anna Spreafico, Kim Ma, Petr Kavan, Gerald Batist, April A. N. Rose

**Affiliations:** 1Gerald Bronfman Department of Oncology, McGill University, Montreal, QC H4A 3T2, Canada; suzanne.kazandjian@mail.mcgill.ca (S.K.); kim.ma.ccomtl@ssss.gouv.qc.ca (K.M.); petr.kavan@mcgill.ca (P.K.); gerald.batist@mcgill.ca (G.B.); 2Segal Cancer Centre, Jewish General Hospital, Montreal, QC H3T 1E2, Canada; 3Lady Davis Institute, Jewish General Hospital, Montreal, QC H3T 1E2, Canada; emmanuelle.rousselle@mail.mcgill.ca (E.R.); matthew.dankner@mail.mcgill.ca (M.D.); 4Division of Experimental Medicine, McGill University, Montreal, QC H4A 3J1, Canada; 5Faculty of Medicine and Health Sciences, McGill University, Montreal, QC H3G 2M1, Canada; 6Rosalind and Morris Goodman Cancer Institute, McGill University, Montreal, QC H3A 1A3, Canada; 7Department of Medical Oncology and Hematology, Princess Margaret Cancer Center, Toronto, ON M5G 2M9, Canada; dave.cescon@uhn.ca (D.W.C.); anna.spreafico@uhn.ca (A.S.)

**Keywords:** BRAF mutation, melanoma, colorectal cancer, non-small cell lung cancer, MAPK pathway

## Abstract

**Simple Summary:**

BRAF mutations are classified into four categories based on molecular characteristics, but only Class 1 BRAF V600 have effective targeted treatment strategies. With increasing access to next-generation sequencing, oncologists are more frequently uncovering non-V600 BRAF mutations, where there remains a scarcity of effective therapies. Responsiveness to MAPK pathway inhibitors differs according to the BRAF mutation class and primary tumor type. For this reason, we sought to determine whether key demographic, genomic, and transcriptomic differences existed between classes. This cross-sectional study analyzes the largest dataset of BRAF-mutated cancers to date. Our findings propose insights to optimize clinical trial design and patient selection in the pursuit of developing effective treatment strategies for patients whose tumors harbor non-V600 BRAF mutations. This study also offers insights into the potential of targeting alternative pathways in addition to the MAPK pathway as part of combinatorial treatment strategies.

**Abstract:**

Background: BRAF mutations are classified into four molecularly distinct groups, and Class 1 (V600) mutant tumors are treated with targeted therapies. Effective treatment has not been established for Class 2/3 or BRAF Fusions. We investigated whether BRAF mutation class differed according to clinical, genomic, and transcriptomic variables in cancer patients. Methods: Using the AACR GENIE (v.12) cancer database, the distribution of BRAF mutation class in adult cancer patients was analyzed according to sex, age, primary race, and tumor type. Genomic alteration data and transcriptomic analysis was performed using The Cancer Genome Atlas. Results: BRAF mutations were identified in 9515 (6.2%) samples among 153,834, with melanoma (31%), CRC (20.7%), and NSCLC (13.9%) being the most frequent cancer types. Class 1 harbored co-mutations outside of the MAPK pathway (TERT, RFN43) vs. Class 2/3 mutations (RAS, NF1). Across all tumor types, Class 2/3 were enriched for alterations in genes involved in UV response and WNT/β-catenin. Pathway analysis revealed enrichment of WNT/β-catenin and Hedgehog signaling in non-V600 mutated CRC. Males had a higher proportion of Class 3 mutations vs. females (17.4% vs. 12.3% q = 0.003). Non-V600 mutations were generally more common in older patients (aged 60+) vs. younger (38% vs. 15% *p* < 0.0001), except in CRC (15% vs. 30% q = 0.0001). Black race was associated with non-V600 BRAF alterations (OR: 1.58; *p* < 0.0001). Conclusions: Class 2/3 BRAFs are more present in Black male patients with co-mutations outside of the MAPK pathway, likely requiring additional oncogenic input for tumorigenesis. Improving access to NGS and trial enrollment will help the development of targeted therapies for non-V600 BRAF mutations.

## 1. Introduction

BRAF is a serine/threonine kinase and a key signaling molecule within the mitogen-activated protein kinase (MAPK) pathway. The MAPK pathway transmits extracellular mitogenic signals to the nucleus of receptive cells, promoting cellular survival and proliferation [1]. BRAF genomic alterations are common in many cancer types and have proven to be potent oncogenic drivers [2].

Oncogenic BRAF alterations are categorized into four distinct categories based on kinase activity, RAS-dependency, and dimerization requirements [1,3,4]. Class 1 BRAF mutants occur at the V600 codon, signal as monomers independent of upstream RAS activation, and exhibit substantially increased kinase activity [5]. Oncogenic non-V600 BRAF mutants hyper-activate the MAPK pathway by forming RAS-independent dimers with intermediate to high kinase activity (Class 2) and RAS-dependent dimers with impaired kinase activity or dead kinase, known as the Class 3 mutations [3,6]. BRAF Fusions function as RAS-independent obligate dimers that signal similarly to Class 2 BRAF mutations.

Class 1 BRAF mutant cancers are targetable with combinations of BRAF, MEK, and EGFR targeted therapies for advanced-stage melanoma, non-small cell lung cancer (NSCLC), and colorectal cancer (CRC) [7,8,9,10,11]. Recently, the FDA granted tumor-type agnostic approval of dabrafenib and trametinib, BRAF and MEK inhibitors, for the treatment of any metastatic BRAF V600E mutant cancer [12]. Conversely, no targeted therapy treatments have been approved for cancers with Class 2 and 3 non-V600 BRAF mutations or BRAF fusions—which comprise approximately 35% of all oncogenic BRAF alterations in adult solid tumors [1]. Clinical data suggest that some patients with Class 2/3 BRAF mutations benefit from MAPK pathway inhibitors, but the response rates are lower than in patients with Class 1 BRAF mutations [13,14].

Multiple mechanisms of acquired resistance to BRAF +/− MEK inhibitors have been described, including via oncogenic co-mutations in genes such as PTEN, NRAS, NF1, and AKT [15]. Treatment response to BRAF and MEK inhibitors has also been shown to vary according to sex, with women deriving more benefit from targeted therapies than men [10]. However, these interactions within the BRAF mutation class across multiple tumor types are not well understood.

We sought to identify clinical, genomic, and transcriptomic variables that could explain the differences in responsiveness to MAPK inhibitors between BRAF mutation classes. We studied the AACR Project GENIE and The Cancer Genome Atlas (TCGA) datasets, a comprehensive cancer genomics database with clinical and next-generation sequencing (NGS) data available from over 150,000 tumor samples [16,17]. Our analysis provides insight into the molecular mechanisms underlying the tumorigenesis of non-V600 BRAF mutant tumors and identifies subgroups of patients most likely to benefit from novel therapeutic approaches.

## 2. Materials and Methods

**Search Strategy:** Using the AACR Project GENIE (v12) cancer database, we analyzed the incidence and distribution of BRAF mutation class in cancer patients according to sex, age, race, sample type, tumor type, and co-occurring mutations [16,17]. Samples with a BRAF mutation and patient age above 20 were included. Patients with missing information on sex, age, unclassifiable BRAF mutations, and VUS mutations (variant of uncertain significance) were excluded (Appendix A). The remaining samples were classified into four categories according to previously published criteria. The incidence of specific BRAF alterations is included in Appendix A [1,18,19].

**Statistical Analysis:** Statistical analysis was performed using the chi-square test and values were corrected using the Benjamini–Hochberg method. Q < 0.05 and *p* < 0.05 were considered statistically significant. For univariable and multivariable analyses of variables associated with V600 vs. non-V600 BRAF mutation status, odds ratio and 95% confidence intervals (CI) were calculated using a probit logistic regression model. All variables with *p* < 0.05 were included in the initial multivariable model. Only variables with *p* < 0.05 in multivariable analysis were included in the final multivariable model. Continuous variables were assessed using the two-way ANOVA test.

**Genomic Analysis:** We compared genomic alterations in tumors with Class 1, 2, or 3 BRAF mutations and in primary vs. metastatic tumor samples. For comparison, according to BRAF Class, the top 30 most frequently co-occurring mutated genes are highlighted in the oncoprint figure. These genes were selected amongst the genes that were significantly differentially altered (q < 0.05) according to BRAF mutation class after excluding genes that were altered in fewer than 15 samples across all classes. Pathway analysis was performed using MSigDB Hallmark analysis and Enrichr [20,21,22,23] on the list of genes that were significantly (q < 0.05) differentially altered between Class 1 vs. Class 2 or Class 1 vs. Class 3. Next, we compared genes that were significantly differentially altered (q < 0.05) between metastatic and primary tumor samples within 9 sub-groups: melanoma; colorectal cancer; or NSCLC with Class 1, 2, or 3 BRAF mutations. All genes that were significantly differentially altered (*p* < 0.05) between metastatic and primary tumor samples and altered in 5 or more samples are depicted, except any genes that were altered in 0% of either primary or metastatic tumors.

**Transcriptomic Analysis:** RNA sequencing data for BRAF mutant melanoma, NSCLC, and CRC cancers were obtained from the TCGA database (195, 31, and 53 total BRAF mutant samples, respectively) [19]. The distribution of samples by BRAF Class is presented in Appendix A section. We performed Gene Set Enrichment Analysis (GSEA) after comparing the tumors with BRAF V600 (Class 1) mutations to tumors with non-V600 BRAF mutations [20,21,24]. Significantly differentially expressed genes that were included in heatmaps and pathway analyses were those that met the following criteria: Base Mean > 50, absolute fold change > 2, and Padj < 0.01.

## 3. Results

### 3.1. Characteristics of BRAF Mutant Patient Cohort

BRAF alterations were identified in 6.2% of samples (Appendix A). Among the classifiable mutations, 3358 (65.6%) were Class 1 mutants, 782 (15.3%) were Class 2 mutants, 759 (14.9%) were Class 3 mutations, and 221 (4.3%) were BRAF Fusions. The most frequent BRAF altered cancer types were melanoma (*n* = 1591), CRC (*n* = 1061), and NSCLC (*n* = 714) (Table 1).

### 3.2. Relationship between BRAF Mutation Class and Co-Occurring Genomic Alterations

Across all BRAF mutant cancers, 228 genes were significantly differentially altered in BRAF Class 1/2/3 mutant tumors (Appendix A). The most frequently differentially altered genes according to BRAF mutation Class are indicated in Figure 1A,B (Appendix A). We validated previously published observations indicating that KRAS, NRAS, and NF1 mutations were more common in Class 2 and 3 mutant tumors vs. Class 1 mutant tumors [6,10,22]. However, we also report several novel gene alterations that differ according to BRAF Class (TERT, TP53, APC, and PIK3CA). Most of these gene alterations were more common in Class 2 and 3 BRAF mutant tumors, but TERT and RNF43 alterations were more common in Class 1 BRAF mutant tumors. We also examined the relationship between the BRAF mutation class and co-occurring genomic alterations within specific cancer types. Of note, most co-occurring mutations in melanoma BRAF Class 2 and 3 (KMT2A, NOTCH1, APC) are of unknown significance, whereas those in Class 1 are amplifications and putative drivers such as NOTCH2 and MET amplifications (Appendix A). In CRC, Class 1 samples had a higher co-mutation burden compared to other BRAF classes (Appendix A). RNF43 truncating mutations most frequently occurred in BRAF Class 1 CRC vs. Classes 2 and 3. Conversely, RAS isomers (KRAS and NRAS) were more likely to be co-mutated in BRAF Class 2 and 3 CRC. In NSCLC, TP53 and SETD2 co-mutations were more frequent in Class 1 (Appendix A). Class 2 and 3 BRAF mutant NSCLC were more likely to have co-occurring mutations in STK11, KEAP1, and KRAS. Across all three cancer types, the mutational landscape of BRAF Fusions was most similar to tumors with Class 1 BRAF mutations.

Next, significantly differentially altered genes (BRAF Class 1 vs. BRAF Class 2/3) were used to perform pathway enrichment analysis. Notch Signaling, UV Response, TGF-Beta Signaling, Wnt-beta Catenin Signaling, and PI3K/AKT/mTOR Signaling pathways were differentially altered between Class 3 and Class 1 tumors (Figure 1C). Within melanoma, Apical Junction, Wnt-beta Catenin Signaling, and UV Response pathways were significantly altered between either Class 2 or Class 3 BRAF mutant tumors. Several additional pathways were significantly altered only in Class 3 melanomas—including E2F targets and G2M Checkpoint (Appendix A). Pathways enriched between Class 3 vs. Class 1 BRAF mutant CRCs and Class 3 vs. Class 1 BRAF mutant melanomas were overlapping (Appendix A). Pathway analysis of altered genes in NSCLC did not yield any significant findings (Appendix A). Overall, across multiple tumor types, Class 2 and 3 BRAF mutant tumors were enriched for alterations in genes involved in ultraviolet (UV) response and Wnt-beta Catenin signaling. Notch Signaling, E2F targets, G2M Checkpoint, and Hedgehog Signaling pathways were also enriched in BRAF Class 3 tumors. We also performed a gene ontology analysis for gene alterations enriched in Class 2 vs. Class 1 and Class 3 vs. Class 1 within all cancer types (Appendix A). Gene alterations of the Class 3 tumors play important roles in the regulation of apoptotic processes and processes regulating cell proliferation and migration, such as the Wnt, mTORC, and Notch pathways (Figure 1C and Appendix A).

### 3.3. Gene Alteration Trends within Primary vs. Metastatic Tumors and BRAF Mutation Class

Next, we sought to identify gene alterations that were enriched in metastatic vs. primary tumors for each BRAF mutation Class in the 3 cancer types where BRAF mutations are most frequently identified: melanoma, colorectal, and NSCLC. In melanoma, TERT promoter mutations, MITF, and CTNNB1 were amongst the genes enriched in metastatic vs. primary Class 1 BRAF mutant tumors (Figure 2A). TERT promoter mutations were also enriched in Class 2 but not in Class 3 BRAF mutant metastatic tumors (Figure 2B,C). Indeed, we did not identify any genes that were significantly enriched in Class 3 BRAF mutant metastatic vs. primary melanoma tumors. Interestingly, metastatic CRC tumors have significantly different mutational landscapes than the primary CRC tumors when the BRAF mutation is Class 1 or 3 but not Class 2 (Figure 2D–F). Our genomic analysis of RAS isoform mutations in BRAF mutant CRCs showed enrichment in the non-V600 BRAF mutants, and in Figure 2F, we show that NRAS mutations are significantly enriched in BRAF Class 3 metastatic vs. primary CRC, whereas KRAS mutations are enriched in BRAF Class 3 primary CRC tumors (*p* = 0.0372 and *p* = 0.0109, respectively). Furthermore, CD274 alterations (commonly known as PDL1) are enriched in BRAF Class 1 metastatic CRC (*p* = 0.0388). Within NSCLC, there were no gene alterations that were significantly enriched in primary tumors, irrespective of BRAF mutation Class, but all three classes of BRAF mutant metastatic NSCLC tumors were enriched for TP53 mutations (*p* < 0.0001) (Figure 2G–I). Furthermore, all three classes of metastatic BRAF mutant NSCLC were significantly enriched for alterations in genes encoding receptor tyrosine kinases, including EGFR and ALK in Class 1; EGFR, ERBB2, ERBB4, FGFR2, and PDGFRA in Class 2; and ERBB3 and FLT3 in Class 3 (Figure 2G–I).

### 3.4. Relationship between BRAF Mutation Class and Gene Expression

To validate the pathway enrichment findings from our genomic analysis at the transcriptional level, we performed GSEA on RNA sequencing data from BRAF mutant melanoma, NSCLC, and CRC. The top 10 gene sets enriched amongst the genes differentially expressed between non-V600 vs. V600 BRAF mutant cancers for each cancer type are indicated in Figure 3. In melanoma, we observed an overlap for gene signatures enriched in Class 2/3 BRAF mutant tumors at the transcriptional and genomic level—these included PI3K/AKT Signaling, Estrogen Response (Early and Late), Mitotic Spindle, G2M Checkpoint, and UV Response Dn (Figure 3 and Appendix A). Pathway analysis of differentially expressed genes between non-V600 and V600 BRAF mutant CRC revealed enrichment of Wnt-Beta Catenin and Hedgehog Signaling pathways in the non-V600 BRAF mutants. Conversely, the enrichment of Apical Junction in the NSCLC non-V600 BRAF mutants was the only commonality between the genomic and transcriptomic analyses in NSCLC.

### 3.5. Relationship between Patients’ Sex and BRAF Mutation Class

The incidence of specific oncogenic genomic alterations varies significantly according to sex [23,25]. Our dataset’s findings confirm that this pattern holds in the distribution of BRAF mutations, with significant variation according to sex. Across all cancer types, males had a higher proportion of Class 3 mutations vs. females (17.4% vs. 12.3% q = 0.003) (Figure 4A). Melanoma, CRC, and NSCLC also exhibited a higher proportion of Class 3 BRAF mutations in males (Figure 4A). Conversely, females had a higher proportion of Class 1 mutations compared to males across all cancer types (69% vs. 62%) and within specific cancer types, including melanoma, CRC, and NSCLC. The relationship between BRAF mutation class and sex was independent of other key clinical and genomic variables in multivariable analysis (OR 1.55 *p* < 0.0001) (Table 2). 

### 3.6. Relationship between Age and BRAF Mutation Class

The age distribution of patients also varied according to the BRAF Class (Figure 4B). The majority (57.5%) of patients included in this analysis were 60 years or older (Table 1). Across all cancer types, the relative proportion of non-V600 BRAF mutations (Class 2, 3, and Fusions) was higher in older patients (age 60+) compared to younger patients (age < 60) (*p* < 0.0001) (Figure 4B and Appendix A). This relationship between BRAF class and age was independent of other clinical or genomic variables—such as RAS mutation status—in multivariable analyses and was observed in melanoma and other cancer types (Table 2). However, in NSCLC and CRC, non-V600 BRAF mutations were more common in younger patients (Figure 4B and Appendix A). CRC with non-V600 BRAF mutations were more frequent in patients who were younger than 50 (34.1% of all BRAF mutations) compared to only 17.9% in patients older than 50 (*p* < 0.0001) (Appendix A).

### 3.7. Relationship between Primary Race and BRAF Mutation Class

The distribution of the BRAF mutation class varied according to the patient’s primary race (Figure 4C). Across all cancers, and specifically within CRCs and other cancer types, Black patients had a lower proportion of Class 1 mutations compared to patients of other races. In NSCLC, Asian patients were less likely to have Class 3 mutations (16%) and more likely to have BRAF fusions (19%) compared to Black patients (32% Class 3 and 4% Fusions) or White patients (30% Class 3 and 3% Fusions). In multivariable analysis, adjusting for primary tumor type, Black race was independently associated with increased odds of having a non-V600 BRAF alteration (OR: 1.58; 95% CI 1.13 to 2.20; *p* < 0.0001) (Table 2).

## 4. Discussion

Effective targeted therapy strategies have not yet been established for cancers with non-V600 BRAF mutations [7,8,9,10,26]. Although preclinical evidence suggests BRAF + MEK inhibitors may be effective in subsets of Class 2 and 3 BRAF mutant tumors, clinical evidence to date indicates that only a small minority of these patients derive meaningful clinical benefit from the same MAPK inhibitors that are highly effective for Class 1 BRAF mutant tumors [13,14,27,28]. Our cross-sectional analysis highlights potential explanations for the lackluster efficacy of MAPK inhibitors in tumors with oncogenic, non-V600 BRAF mutations compared to those with Class 1 V600 BRAF mutations.

Our genomic analysis highlights the cooperation of mutations outside of the MAPK pathway, such as TERT or RNF43, in Class 1 BRAF mutant tumors, while Class 2 and Class 3 BRAF mutations co-exist with additional MAPK pathway mutations. These genes include RAS (KRAS, HRAS, NRAS), NF1, and genes encoding receptor tyrosine kinases, such as EGFR, ERBB2, MET, and RET. This finding is consistent with the less potent activation of downstream MAPK pathway activation in non-V600 Class 2 and 3 BRAF mutations and highlights additional oncogenic inputs are required to cooperate with Class 2 and 3 BRAF mutations to elicit sufficient MAPK signaling output to promote tumorigenesis [3,4].

Significant differences were observed in the genomic landscape across three common cancer types with a high proportion of BRAF mutations: melanoma, NSCLC, and CRC. Non-V600 BRAF mutant NSCLC tumors co-exist with mutations of the MAPK pathway signaling proteins, such as KRAS, but also frequently with loss of function (LoF) mutations in tumor suppressor genes involved in cellular processes such as metabolism (STK11) and oxidative stress response (KEAP1). STK11 is a kinase acting as a metabolic sensor and, when mutated, increases mTOR activity and cell proliferation [29]. Nearly 30% of NSCLC with Class 2/3 BRAF mutations had co-occurring LoF STK11 mutations. These findings suggest that Class 2 and 3 mutations—which have a lower transformation capacity than Class 1 BRAF mutations—may cooperate with other signaling pathways to drive tumorigenesis [2]. Notably, our genomic alteration in metastatic tumor data shows that KEAP1 mutations are enriched in BRAF Class 2 NSCLCs, suggesting that cooperation between Class 2 BRAF mutations and KEAP1 loss-of-function mutations may promote cancer progression. Therapeutic strategies targeting mTOR may cooperate with MAPK inhibitors in NSCLC. KEAP1 and STK11 mutations are negative prognostic factors in NSCLC [30]. Novel agents like mTORC 1/2 inhibitors (Sapanisertib) alone or in combination with Telaglenastat (glutaminase inhibitor) are being studied in KEAP1/STK11 mutant NSCLC (NCT03872427; NCT04250545) [31].

In melanoma, we found that BRAF Class 1 mutations occur in younger patients compared to Class 2/3 non-V600 BRAF mutations. BRAF mutant melanoma cancers carry a high co-mutation burden (of which many gene alterations are of unknown significance). UV Response pathways were significantly enriched in Class 2/3 mutants vs. BRAF Class 1 melanomas, which could explain the increased tumor mutation burden in these tumors. Wnt signaling drives transformation and proliferation in melanoma [32]. We found a significant enrichment of alterations of this pathway in Class 2/3 melanomas compared to Class 1. Wnt-beta catenin pathway was also enriched in NSCLC and CRC with Class 2 or 3 BRAF mutations. Alterations in G2M Checkpoint and E2F targets (which include genes critical for cell cycle progression) were also enriched in BRAF Class 3 melanoma, indicating dysregulation of the cell cycle in these tumors. Together, these data highlight the potential for inhibitors of Wnt signaling and cell cycle progression (i.e., CDK4/6 inhibitors) to be used for the treatment of Class 2/3 BRAF mutant cancers.

In our analysis, alterations in genes regulating the SWI/SNF chromatin remodeling complex (LoF ARID2, LoF ARID1A, and SMARCA4) were most frequent in Class 3 melanomas. LoF ARID1A mutations independent of other mutations are not sufficient for tumorigenesis but may accelerate tumor development driven by co-occurring oncogenes [33]. ARID2 depletion leads to transcription changes in genes regulating melanoma metastasis through BAF redistribution [34]. Thus, the relatively high incidence of co-occurring alterations of SWI/SNF complex-related genes may represent a novel therapeutic opportunity in Class 3 BRAF mutant tumors. Current preclinical data support the use of PARP inhibitors, Aurora kinase inhibitors, and SMARCA2 degraders in tumors with ARID1A, ARID2, and SMARCA4 LoF mutations [35,36]. These inhibitors may also warrant further investigation in subsets of Class 3 BRAF mutant melanoma.

In CRC, we observe similar frequencies of co-existing RAS mutations in Class 2 and Class 3 mutant tumors. Interestingly, metastatic Class 3 CRC tumors were enriched for NRAS mutations but not KRAS mutations compared to Class 3 primary CRC tumors. It has previously been reported that Class 1 BRAF mutations commonly occur in cancers that arise in the right colon, whereas Class 2 and 3 non-V600 BRAF mutations more commonly arise in the left colon [37]. Importantly, there are different embryological origins of the cells that give rise to tumors in the left and right colon (hind-gut vs. mid-gut derivatives) [38]. Right-sided CRC frequently harbors Class 1 BRAF mutations, a finding that is compatible with the fact that tumors deriving from this tissue are largely driven by EGFR-independent oncogenic inputs [14]. Meanwhile, left-sided disease relies upon EGFR as a critical driver of cell proliferation through the MAPK pathway, and therefore this EGFR signaling is amplified by additional downstream MAPK driver mutations, such as Class 2 and 3 BRAF mutations. In retrospective data, it appears that there is a better response rate to EGFR inhibitors in mCRC with Class 3 vs. Class 2 BRAF mutations [39]. Targeted therapy is a second-line option after failure of chemotherapy for metastatic CRC harboring Class 1 mutation [9]. However, it is not yet known whether the targeted therapy combination of BRAF + EGFR inhibitors that are effective for Class 1 BRAF mutant mCRC are also effective in Class 2 and 3 BRAF mutant mCRC—but ongoing clinical trials are actively investigating this question [9,40].

We identified a strong relationship between Black race and non-V600 BRAF mutations, especially in CRC, where these mutations occur in younger patients. The median age (at the time of sequencing) for patients with Class 2 and 3 BRAF mutant CRC was younger than Class 1. The incidence of early onset CRC has been increasing and is even more predominant in the Black community [41]. Despite a lower incidence of BRAF V600 mutated CRC in Black patients, the mortality for this population remains high, as they tend to present with metastatic disease [25,41]. APC mutations occur more commonly in early onset CRC and are associated with a poor prognosis [42]. We observed a high incidence of co-occurring APC mutations in Class 2/3 BRAF mutant CRC compared to Class 1. Nevertheless, truncating driver APC mutations are enriched in metastatic Class 1 CRC vs. primary Class 1 CRC, which highlights the importance of these loss-of-function mutations in colorectal cancer progression. APC mutations result in increased Wnt-beta Catenin pro-oncogenic signaling, which we found to be enriched overall in CRC Class 3 vs. Class 1 BRAF mutant tumors [43]. Black and other non-white patients are typically under-represented in cancer sequencing datasets and in clinical trials in the oncology [44]. Clinicians should have a high index of suspicion for the presence of non-V600 BRAF mutations in younger, Black patients with metastatic CRC. These are potentially actionable mutations for which treatment strategies have not yet been defined.

Finally, we found that BRAF Class 3 mutations were more common in men than women. Men with melanoma have worse outcomes than women [45]. Moreover, men with Class 1 BRAF mutant melanoma are less likely to benefit from BRAF + MEK inhibitor therapy [46]. This finding suggests there may be a hormonal explanation for these differences. Supplemental testosterone mitigates the effects of BRAF + MEK inhibition, and blockade of the androgen receptor (AR) promotes the anti-tumor activity of BRAF + MEK inhibitors in BRAF Class 1 mutant melanoma mouse models [46]. Class 3 BRAF mutant tumors rely on upstream AR signaling for oncogenic MAPK pathway activation [47]. The increased incidence of Class 3 BRAF mutations in men and the limited efficacy of MAPK inhibitors in these tumors suggests a role for investigating inhibitors of AR activity in men with Class 3 BRAF mutant tumors [14]. Similar to mCRC, targeted therapy is the preferred second-line option for metastatic V600E mutated melanoma, given that longer response rates are observed with combination immunotherapy [48]. Sequencing with anti-PD1/PD-L1 first heightens the response to targeted therapy by promoting a pro-inflammatory response of macrophages [49].

Our data have several limitations. The age reported is the age at the time of tumor sequencing, which may have occurred years following diagnosis. Primary race is self-reported, and there are insufficient data for patients with mixed-race backgrounds. Biological and health outcomes associated with gender or race are a consequence of differences in environmental exposures, diet, health care access, systemic and structural racism—rather than genetic differences between individuals of different races [50]. Information on social health determinants and lifestyle factors, known to increase tumorigenesis, is lacking. Skin pigmentation, known to influence the incidence of melanoma, as well as phenotyping information, is lacking, which limits the interpretation of our results [51]. Samples were sequenced with various assays; therefore, no analysis was performed on genes sequenced in a minority of samples. Results from genomic pathway enrichment analyses should be considered hypothesis-generating as the functional significance of gene alterations was not always known. We validated the results from the genomic pathway enrichment with the standard approach of transcriptional analysis and gene set enrichment analysis, giving us confidence in our conclusions. Nonetheless, further preclinical work is required to bridge the gap between our research and the development of novel therapeutic approaches based on our identification of potentially actionable pathways in non-V600 BRAF mutant tumors.

## 5. Conclusions

In conclusion, our results may suggest that non-V600 BRAF mutations occur more commonly in Black patients, older age, NSCLC and co-mutations within the MAPK pathway. Class 3 BRAF mutations are more common in men raising the possibility of non-genomic signaling from androgen receptor pathway. More research is needed to determine the molecular mechanisms governing these associations. Our analysis of a large cohort of BRAF mutant cancers will help identify subsets of populations that would benefit from novel targeted therapies. It underlines the importance of continuing cancer sequencing projects and the insight that can be gained from these large-scale sequencing efforts—particularly for rare driver mutations. 

## Figures and Tables

**Figure 1 cancers-16-00445-f001:**
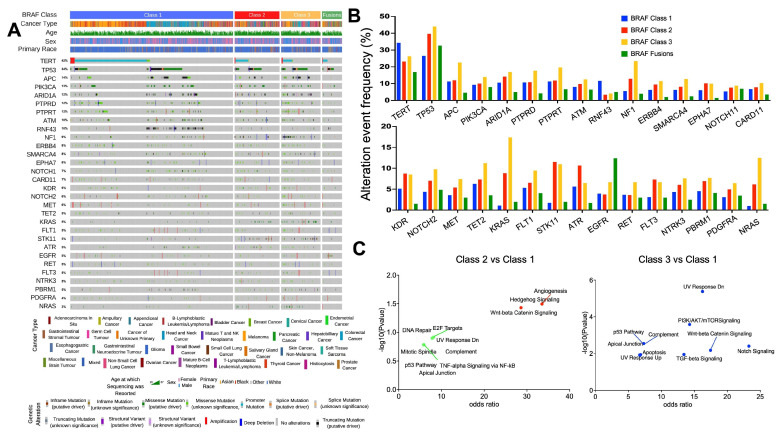
The genomic landscape of BRAF mutant tumors. (**A**) Oncoprint highlighting the top 30 most frequent genes that are differentially altered between tumors with Class 1/2/3 BRAF mutations. (**B**) Histogram highlighting the incidence of gene alterations within each BRAF class. (**C**) The filtered list of genes that were significantly differentially altered according to BRAF Class 1/2 and 1/3 status across all cancers (*n* = 18 and *n* = 59, respectively, Q < 0.05) was subjected to pathway analysis using the MSigDB Hallmark algorithm. Pathways that were over-represented in this list of genes are indicated in blue, red, and green (*p* < 0.05 and Q < 0.05, *p* < 0.05, Q < 0.2, and *p* < 0.2 and Q < 0.2, respectively).

**Figure 2 cancers-16-00445-f002:**
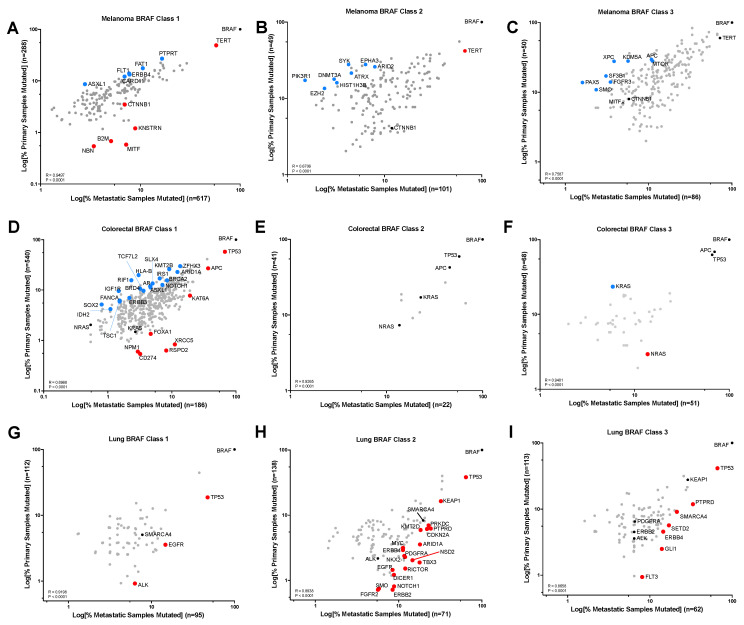
Gene alterations enriched in primary vs. metastatic BRAF mutant tumors. Gene alterations present in melanoma (**A**–**C**), CRC (**D**–**F**), and NSCLC (**G**–**I**) based on percent of samples with the gene alteration in primary (y-axis) or metastatic (x-axis) tumors. Genes that were present in a minimum of 1 primary and metastatic tumor are included. Pearson Correlation (R) was calculated for each panel. Colored dots represent significant enrichment (*p* < 0.05) in primary (blue) or metastatic (red) tumors (*p* value derived from two-sided Fisher Exact test).

**Figure 3 cancers-16-00445-f003:**
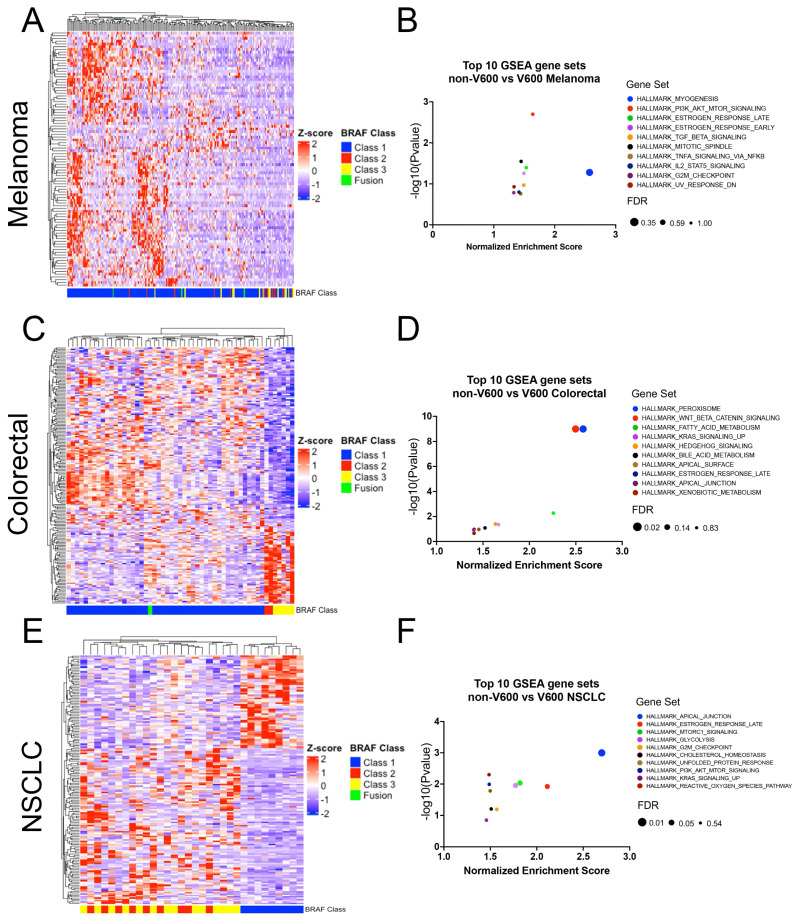
The transcriptomic landscape of BRAF mutant tumors. (**A**) Heatmap of the differentially expressed genes (*n* = 93) in Class 1 (V600) vs. Class 2/3/Fusion (non-V600) BRAF mutant melanoma tumors (TCGA, *n* = 195 samples). (**B**) Top 10 GSEA gene sets enriched in non-V600 BRAF mutant melanoma tumors. (**C**) Heatmap of the differentially expressed genes (*n* = 182) in BRAF mutant V600 vs. non-V600 colorectal tumors (TCGA, *n* = 53 samples). (**D**) Top 10 GSEA gene sets enriched in non-V600 BRAF mutant colorectal tumors. (**E**) Heatmap of the differentially expressed genes (*n* = 160) in V600 vs. non-V600 BRAF mutant NSCLC tumors (TCGA, *n* = 32 samples). (**F**) Top 10 GSEA gene sets enriched in non-V600 BRAF mutant NSCLC tumors. Genes plotted for each heatmap can be found in Appendix A.

**Figure 4 cancers-16-00445-f004:**
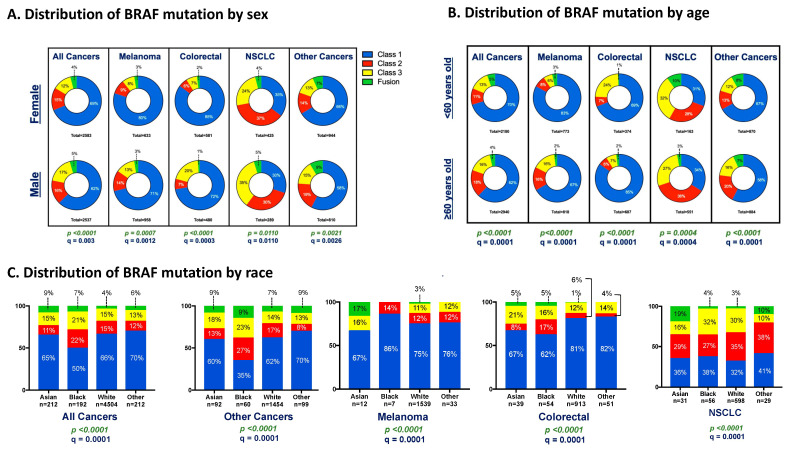
Distribution of BRAF mutation class according to sex, age, and race across cancer type. The frequency of each BRAF class is shown in subgroups defined by sex (**A**), age (**B**), and primary race (**C**) among all patients, melanoma, CRC, NSCLC, and all other cancers (all patients excluding melanoma, CRC, and NSCLC). Values shown within each category represent the proportion of patients expressing each BRAF Class within cancer types according to sex (**A**), age (**B**), and primary race (**C**). *p*-value was calculated through the chi-square test for each contingency table and was then corrected using the Benjamini–Hochberg method to determine false discovery rate–corrected q value, which was considered significant when q was less than 0.05.

**Table 1 cancers-16-00445-t001:** Baseline demographic and clinical characteristics of patients.

Characteristics	All Patients (*n* = 5120)	Class 1 (*n* = 3358)	Class 2 (*n* = 782)	Class 3 (*n* = 759)	Fusion (*n* = 221)
**Age**					
Median—yr	62 (20–90)	62 (20–90)	65 (20–90)	64 (22–90)	59 (20–90)
Distribution—No./total No. (%)					
<60 yr	2180 (43)	1532 (70)	241 (11)	291 (13)	116 (5)
≥60 yr	2940 (57)	1826 (62)	541 (18)	468 (16)	105 (4)
**Gender—No./total No. (%)**					
Male	2537 (50)	1581 (62)	400 (16)	441 (17)	115 (5)
Female	2583 (50)	1777 (69)	382 (15)	318 (12)	106 (4)
**Race—No./total No. (%)**					
Asian	212 (4)	138 (65)	24 (11)	32 (15)	18 (9)
Black	192 (4)	95 (50)	42 (22)	41 (21)	14 (7)
White	4504 (88)	2977 (66)	691 (15)	659 (15)	177 (4)
Other	212 (4)	148 (70)	25 (12)	27 (13)	12 (6)
**Sample type—No./total No. (%)**					
Primary site	2421 (47)	1560 (64)	387 (16)	369 (15)	105 (4)
Metastatic	2122 (41)	1379 (65)	332 (16)	304 (14)	107 (5)
Not reported	577 (11)	419 (73)	63 (11)	86 (15)	9 (2)
**Type of Tumour—No./total No. (%)**					
Melanoma	1591 (31)	1187 (35)	189 (24)	174 (23)	41 (19)
Colorectal cancer	1061 (21)	842 (25)	67 (9)	136 (18)	16 (7)
Non-small cell lung cancer	714 (14)	237 (7)	243 (31)	203 (27)	31 (14)
Thyroid cancer	689 (12)	656 (20)	14 (2)	0	19 (9)
Glioma	201 (4)	118 (4)	26 (3)	23 (3)	34 (15)
Unknown primary	125 (2)	56 (2)	25 (3)	38 (5)	6 (3)
Bladder cancer	39 (1)	3 (0.1)	16 (2)	18 (2)	2 (1)
Hepatobiliary	75 (1)	27 (0.8)	20 (3)	23 (3)	5 (2)
Pancreatic cancer	72 (1)	25 (0.7)	31 (4)	5 (1)	11 (5)
Prostate cancer	72 (1)	1 (0.03)	42 (5)	6 (1)	22 (10)
Other	481 (9)	206 (6)	108 (14)	133 (18)	34 (15)
**Co-Mutations—No./total No. (%)**					
NF-1	234 (5)	59 (25)	57 (24)	114 (49)	4 (2)
RAS (HRAS, NRAS, and/or KRAS)	390 (8)	44 (11)	115 (30)	225 (58)	6 (2)
**Variant Allele Frequency (%)**					
<26%	2242 (47)	1446 (64)	382 (17)	414 (18)	-
≥26%	2485 (53)	1802 (73)	364 (15)	319 (13)	-

**Table 2 cancers-16-00445-t002:** Multivariable analyses of factors associated with non-V600 BRAF mutations (Class 2/3/Fusion) vs. V600 BRAF mutations (Class 1) OR = odds ratio; CI = Confidence Interval.

	Univariable			Multivariable		
	OR	95% CI	*p* Value	OR	95% CI	*p* Value
Gender						
Male vs. Female	1.33	1.19–1.50	<0.0001	1.55	1.35–1.76	<0.0001
Age						
≥60 vs. <60	1.44	1.28–1.62	<0.0001	1.28	1.12–1.46	<0.0001
Primary Race						
Black vs. other	2.00	1.50–2.67	<0.0001	1.58	1.13–2.20	0.007
Primary tumour type						
Melanoma	0.54	0.48–0.62	<0.0001	0.50	0.43–0.59	<0.0001
Colorectal	0.42	0.36–0.50	<0.0001	0.38	0.31–0.46	<0.0001
NSCLC	4.89	4.13–5.79	<0.0001	3.08	2.53–3.75	<0.0001
Sample Type						
Metastatic vs. primary	0.97	0.86–1.10	0.699	-	-	-
Genomic co-mutations						
RAS mt vs. wild-type	18.4	13.37–25.34	<0.0001	19.18	13.80–26.64	<0.0001
Variant Allele Frequency						
≥26% vs. <26%	0.68	0.61–0.78	<0.0001	-	-	-

## Data Availability

All data used in this manuscript are publicly available through the cbioportal: https://www.cbioportal.org/. Data was accessed on 30 August 2022.

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
