# Peer review of "The Clinical, Genomic, and Transcriptomic Landscape of BRAF Mutant Cancers"

_cancers, 2024, doi:10.3390/cancers16020445_

Round 1

Reviewer 1 Report

Comments and Suggestions for Authors

Reviewer’s comments

This manuscript which is titled as “The clinical, genomic, and transcriptomic landscape of BRAF mutant cancers”, by Suzanne and colleagues, explores the molecular characteristics of BRAF mutant tumors, emphasizing the lack of established effective treatments for Class 2/3 or BRAF Fusions. The authors investigate whether BRAF mutation class varies based on clinical, genomic, and transcriptomic variables in cancer patients. However, a few points require attention and clarification:

1. Age Discrepancy in CRC: The observation that non-V600 mutations are more common in older patients, except in CRC, is intriguing. Please offer a possible explanation.

2. Genomic Alterations: Provide additional details on specific genomic alterations within each BRAF mutation class, especially those outside the MAPK pathway.

3. How might the observed differences in BRAF mutation classes impact treatment strategies or patient outcomes?

Reviewer 2 Report

Comments and Suggestions for Authors

The manuscript investigates BRAF mutations in various cancers, analyzing data from the AACR Project GENIE (v12) database. The study focuses on mutation classes, incidence, and distribution across various factors. Employing genomic, transcriptomic, and statistical analyses, the study revealed distinct genomic variances among BRAF mutation classes in melanoma, NSCLC, and CRC, unveiling potential avenues for treatment. The authors did further analysis about the involved pathways in each class. They also examined the impact of race and sex. Overall, the study provides valuable insights into the intricate landscape of BRAF mutant cancers, paving the way for potential personalized therapeutic interventions. To enhance the depth and breadth of the findings, the authors should incorporate additional analyses, including:

1) Explore how the mutational landscape may change during disease progression and treatment.

Author Response

The manuscript investigates BRAF mutations in various cancers, analyzing data from the AACR Project GENIE (v12) database. The study focuses on mutation classes, incidence, and distribution across various factors. Employing genomic, transcriptomic, and statistical analyses, the study revealed distinct genomic variances among BRAF mutation classes in melanoma, NSCLC, and CRC, unveiling potential avenues for treatment. The authors did further analysis about the involved pathways in each class. They also examined the impact of race and sex. Overall, the study provides valuable insights into the intricate landscape of BRAF mutant cancers, paving the way for potential personalized therapeutic interventions. To enhance the depth and breadth of the findings, the authors should incorporate additional analyses, including:

1) Explore how the mutational landscape may change during disease progression and treatment.

Thank you for your comments, this is an important avenue to investigate. I have performed further analyses comparing the gene alterations occurring in melanoma, CRC and NSCLC primary and metastatic tumors and separating these results by BRAF mutation class. We have added Figure 2 and described the results in lines 207-228. In terms of disease progression, we have investigated whether or not we observe a difference in incidence of V600 vs Non-V600 in metastatic vs. primary samples and found that there is no association with BRAF mutation class and primary vs. metastatic site of origin of biopsy. We have included this info in the revised table 2 where we show no significant differences in univariable or multivariable analysis. In response to the question about how the mutational landscape may change with disease treatment, we are unable to answer this question with this dataset although it is a very important question. Currently no treatment strategies have been established for cancers with Non-V600 mutation and more research is needed to evaluate emerging therapeutic modalities and new classes of therapies in cancers driven by Non-V600 BRAF mutation.

Reviewer 3 Report

Comments and Suggestions for Authors

I congratulate Suzanne Kazandjian and her co-authors for their extensive work on the landscape of BRAF mutant cancers. This is a very complex field not completely analyzed for its implication in the different tumors and the different classes of mutation.

The first point that should be evaluated is the difference between driver mutations and passenger mutations, especially when different gene mutations are present. Besides that, even when a class I BRAF mutation is identified, as a driver, the effect of BRAF and Mek inhibitors are not the same in the   different tumors, for instance the clinical effect is larger and longer in melanoma than in NSCLC and in colon cancer and other cancers, indicating that something else is acting.

Clinical studies, and Real World Experiences, in melanoma patients with BRAF mutation, concord that also the sequence of treatments influence the survival, favoring immune checkpoint use as first choice followed by target therapy and not the inverse.

A point not considered, is the high variability of the BRAF variant allele frequency (VAF), in some papers ranging from 3% to 90%, and VAF impacts, on multivariate analysis, on PFS and OS in published studies, at least in melanoma patients.

Fig 1 C shows genes that were significantly differentially altered according to BRAF Class 1/2 and 1/3 status, the different pathways involved highlight the complexity of interactions, and the necessity to identify the hierarchical interactions, between the different pathways, to be able to plan clinical trials with a solid biologic background

The authors stated “results from genomic pathway enrichment analyses should be considered hypothesis-generating as the functional significance of gene alterations was not always known” and this aspect should moderate the conclusion that their analysis of a large cohort of BRAF mutant cancers will help identify subsets of 319 populations that would benefit from novel targeted therapies

Actually, we are a lot far from that achievement, the more complex the interactions, the more quickly the tumor could develop resistance to the treatment, however, to understand the biological pathway subtending the failure is of paramount relevance.

I do not share some explanations like “men with Class 1 BRAF mutant melanoma are less likely to benefit from BRAF + MEK inhibitor therapy may be suggesting a hormonal explanation for these differences, since the effect is seen in all ages, and does not explain the better effect in women all ages considered.  Hormonal differences influence also the immunological response, and clinical effects, especially the long-lasting ones, could be related to a better immunological response.

Reviewer 4 Report

Comments and Suggestions for Authors

The manuscript by Kazandjian et al. provides an interesting bioinformatics approach to understand non V600 BRAF mutations better by evaluating their association with various parameters. This is interesting to do and may help understanding which patients’ cancer may be driven by what pathways and possibly help managing patients with class 2/3/fusion mutations in BRAF better.

There are a few issues that the authors should revise in the manuscript or supplementary document:

Major:

11.      The supplementary items seem to simply be placed into a pdf document without clarity. Figures are not labelled (Fig 1, 2 and so on, have no title or Figure legends. It is unclear which table is what (not number or title). The authors do refer to Supp figures/tables although those are not identifiable to the reader.

22.      The race nominators are unclear. Does “black race” refer to Canadians with African origin or is it simply a measure of skin pigmentation, regardless of ethnicity. Similarly, “white” may not be an ideal race definition. These definitions prevent conclusions regarding ethnic/genetic background vs phenotypic appearance and skin pigmentation. Especially in melanoma effects of strong vs fair skin pigmentation are possible, but then for example many individuals that fall under the “Asian” classification may also have very dark skin complexity which could affect interpretation. While the data in public databases may be limited this at least warrants some discussion in the Discussion.

33.      In Table 1 the proportion [%] of class mutations for the various cancer types seems to be related to the total number of mutations across the cancers. Wouldn’t it be more interesting to show what proportion of lets say melanoma are class 1, 2, 3 or fusion?

44.      Figure 1//Figure 2: Did the authors embark on gene ontology mapping to elucidate possible pathways driving the cancers carrying different class BRAF mutations? Would be interesting to see some schematics of pathways and easier for a reader to get their head around compared to just pinpointing selected pathways out in the text.

55.      Figure 2: high resolution images are required the ones provided a too pixelated.

66.      Table 3: I am not sure I understand or the authors have specified which parameters they included in the multivariate analysis presented here.

Minor

77.      Line 72 add “s” to patient

88.      Figure 3 should appear before Table 2 as it is referred to in the text before it.

Round 2

Reviewer 1 Report

Comments and Suggestions for Authors

I have no further comment.